# Evaluation of *rat* pancreatic islets damage after siRNA microporation

**Veronika Tomsovska**[1,2], **Ivan Leontovyc**[1], **Klara Zacharovova**[1], **Eva Fabryova**[1,2], **Tomas Koblas**[1], **Zuzana Berkova**[1☯*], **Jan Kriz**[1☯*]

**1** Institute for Clinical and Experimental Medicine, Prague, Czech Republic, **2** Charles University, First Faculty of Medicine, Prague, Czech Republic

☯ These authors contributed equally to this work.
* jan.kriz@ikem.cz (JK); zuzana.berkova@ikem.cz (ZB)

## Abstract

Small interfering RNA (siRNA) can be used for the temporary inhibition of gene expression, and one possible delivery method is microporation. In this study, we evaluated whether microporation is a suitable technique for the transfection of pancreatic islets and whether the selected siRNA is recognized by islet cells as foreign, potentially triggering an inflammatory or cytotoxic response. Rat islets were transfected with siRNA by microporation 24 hours after isolation. Another 24 hours post-transfection, the islets were assessed for viability, function, and the expression of inflammatory markers. Microporation induced mild but, in some assays, statistically significant stress. Flow cytometry revealed 9.94% dead cells in the negative control, 16.33% in islets microporated without siRNA, and 14.50% in islets microporated with siRNA. These results were confirmed by live/dead staining using Propidium iodide and Acridine orange. In contrast, caspase 3/7 staining, insulin secretion assays, and qRT-PCR analysis of inflammatory markers showed no statistically significant differences between the negative control and microporated groups. In addition, there were no significant differences in any of the assays between islets microporated with or without siRNA, or between the siRNA-only group and the negative control. These findings indicate that the selected siRNA is non-toxic to pancreatic islets and does not elicit an inflammatory response. Although microporation induces mild cellular stress and increased cell death, it may still be considered as a possible method for the transfection of pancreatic islets.

## Introduction

Transplantation of isolated pancreatic islets (PI) into the portal vein can be used for the treatment of selected patients with type 1 diabetes mellitus [1–3]. Direct contact with the recipient's blood causes damage to graft cells through nonspecific inflammation and hypoxia due to IBMIR (instant blood-mediated inflammatory

**Data availability statement:** Data are uploaded and will be held in the following public repository: https://doi.org/10.5281/zenodo.17296985.

**Funding:** 1) Supported by the project National Institute for Research of Metabolic and Cardiovascular Diseases (Programme EXCELES, ID Project No. LX22NPO5104) - Funded by the European Union – Next Generation EU. 2) Supported by Ministry of Health, Czech Republic - conceptual development of research organization ("Institute for Clinical and Experimental Medicine – IKEM, IN 00023001") 3) "Supported by the Ministry of Health of the Czech Republic in cooperation with the Czech Health Research Council under project No. NW25-01-00221. All rights reserved".

reaction — activation of complement, activation of the coagulation cascade, platelet aggregation, recruitment of leukocytes), which is triggered and stimulated mainly by tissue factor. Another function of tissue factor is the stimulation of angiogenesis, which is important for the successful connection of the graft to the recipient's blood vessels in the early period after transplantation [4–8]. To minimize early graft damage, it may be beneficial to transiently reduce or suppress tissue factor expression. This can be achieved using RNA interference (RNAi), an evolutionarily conserved cellular mechanism that protects against exogenous nucleic acids [9–12].

Among the different RNAi effectors, we selected small interfering RNA (siRNA) for its specificity and efficiency. These double-stranded molecules, approximately 21–23 nucleotides in length, bind to target mRNA with full sequence complementarity. Once incorporated into the RNA-induced silencing complex (RISC), the siRNA guides the complex to degrade the corresponding mRNA, thereby preventing its translation into protein. This post-transcriptional silencing occurs without altering the nuclear DNA and can be directed toward virtually any gene using artificially designed siRNA sequences [13–15].

The delivery of siRNA towards cells was performed using microporation — a refined electroporation technique that uses a microporation tip. Short electrical pulses are applied to cells, creating temporary pores to the cell membrane allowing nucleic acids to enter the cell cytoplasm [16–19]. Compared to conventional cuvette-based electroporation, microporation offers several advantages: higher transfection efficiency, better post-transfection cell viability, and greater consistency due to reduced electrode surface area. This results in more stable pH and temperature conditions and lower ion release during the procedure. These benefits collectively improve cell survival and transfection outcomes [16,18,20].

Artificially introduced siRNAs can be recognized by the cell as foreign substance and trigger a defensive inflammatory response, commonly referred to as a "nonspecific off-target effect." This effect may entirely overlay the intended purpose of siRNA transfection. A specific off-target effect occurs when siRNA incorrectly binds to an unintended mRNA sequence, thereby silencing a different gene [21–23].

Modification of siRNA nucleotides and structure together [24] with optimization of transfection method and siRNA dosing can reduce the risk of off-target effects [25–28]. The reported study aimed to elucidate the potential side effects of siRNA (anti-tissue factor mRNA), delivered by microporation, on pancreatic islet cells of Rattus norvegicus. Specifically, the influence on cell viability, insulin secretion, and activation of inflammatory response genes were tested.

## Methods

Experiments were approved by The Animal Care Committee of the Institute for Clinical and Experimental Medicine and Ministry of Health of Czech Republic (Permit Number: 12817/2023-7/OVZ).

## Design of the experiment

**Pancreatic islets transfection.** After overnight culture of the isolated pancreatic islets distinct experimental groups were established to evaluate the effects of siRNA delivery and immune stimulation:

A) the negative control group (CTRL-) consisted of islets cultured under standard conditions

B) siRNA-only group (Si) included islets incubated with siRNA without the use of a transfection method.

C) microporation-only group (Mi) was composed of islets subjected to microporation in the absence of siRNA, to assess the effect of the delivery technique alone.

D) microporation with siRNA group (Mi+Si), included islets treated with siRNA delivered via microporation, enabling evaluation of the combined impact of both siRNA and the delivery method.

E) the positive control group (CTRL+) included islets that were microporated and subsequently stimulated with Poly-inosinic:polycytidylic acid Poly(I:C), a synthetic analog of viral double-stranded RNA, to activate an innate immune response and mimics viral infection.

After intervention all PIs were cultured in 12-well plates for 24 hours in a humidified incubator maintained at 37 °C with 5% $CO_2$.

**Evaluation of pancreatic islets after treatment.** After the incubation period, the islets from each group were subjected to the following assays:

i)   **morphological analysis** in the brightfield (light) microscopy

ii)  **Cell viability** using Propidium iodide and Acridine orange staining

iii) detection of apoptotic cells in islets using **caspase 3/7 staining**

iv)  **Glucose stimulated insulin secretion**

v)   **qRT-PCR** expression analysis of IFN-α, IFN-β, and TNF-α

vi)  **Flow cytometry** analysis of apoptotic and dead cells in islets

**Isolation of pancreatic islets.** *Brown Norway (BN) rats* were used as islets donors. Pancreatic islets were isolated from donor's pancreas by collagenase (Collagenase from Clostridium histolyticum, (Merck KGaA, Darmstadt, Germany)) digestion. Then the islets were separated and purified from exocrine tissue by centrifugation in Ficoll gradient (Cytiva, Marlborough, Massachusetts, USA). Islets were counted and divided into 12-well plate of 150–200 PI (corresponds to 300,000–400,000 cells). The islets were cultivated overnight in 12-well plates in CMRL – 1066 (PAN-Biotech GmbH, Aidenbach, Germany) medium supplemented with Fetal Calf Serum 10%, HEPES 5%, Penicillin/Streptomycin 1% and Glutamine 1% (all chemicals from Merck KGaA, Darmstadt, Germany) in a humidified incubator with 5% CO2 atmosphere at 37°C.

**Transfection pancreatic islets. Microporation:** For microporation, Neon™transfection (Thermo Fisher Scientific Inc., Waltham, Massachusetts, USA) system was used and the whole procedure was done in 100 µl microporation tips (Neon™ Transfection System 100 µl Kit). After overnight cultivation, 150–200 islets were hand- picked into an Eppendorf tube and culture medium with antibiotics was replaced with 98 ul of R buffer and 2 ul 100 µM siRNA (Thermo Fisher Scientific Inc., Waltham, Massachusetts, USA, siRNA ID: s130189) for microporation with siRNA, or just 100 µl R buffer for microporation itself, or with 70 µl R buffer and 30 µl of immunostimulant Poly(I:C) (Tocris Bioscience, Bristol, UK, cat. n. 4287, UK) (1 mg/ml) for positive control. This suspension was collected with a 100 µl microporation tip and relocated to an microporation tube with 3 ml of E2 buffer and placed into the microporation system. Microporation was performed by 2 pulses with voltage of 950 V and time of pulse of 30 ms. The suspension in the tip was placed into a well with 900

ul media without antibiotics (total volume in the well was 1 ml). Final concentration of siRNA in the well was 200 nM and, final concentration of Poly(I:C) was 30 ug/ml.

**Cultivation of PI with siRNA, without microporation:** Culture medium with ATB was replaced by medium without ATB, and 2 ul (100 μM) of siRNA was added to final concentration 200 nM.

**Control sample without treatment:** Culture medium with ATB was replaced by medium without ATB. The plate with islets after intervention was placed in a humidified incubator with 5% $CO_2$ atmosphere at 37°C for 24 h.

**Microscopy examination of pancreatic islets.** All microscopic images were acquired using the EVOS FL Auto microscope (Thermo Fisher Scientific Inc., Waltham, Massachusetts, USA). First, morphological analysis was performed under brightfield (light) microscopy. Cell viability was assessed using **Acridine orange** (5 μg/ml) and **Propidium iodide** (100 ug/ml) (Merck KGaA, Darmstadt, Germany).

For the staining, both dyes were mixed in a 1:1 ratio with~50 islets in culture medium in a 24-well plate. After 10 minutes incubation at room temperature with occasional gentle agitation, suspension was diluted with PBS (Merck KGaA, Darmstadt, Germany). Viable cells (stained green by Acridine orange) and non-viable cells (stained red by Propidium iodide) were quantified under fluorescence microscope. Pancreatic islets were categorized based on the approximate percentage of viable cells: 0%, 10%, 25%, 50%, 75%, or 100%. The overall viability was calculated as the average of scores counted by three evaluators and expressed as the percentage of live cells.

To evaluate apoptosis caspase 3/7 staining was performed. PIs were incubated with 5 μl CellEvent™ Caspase 3/7 Detection Reagents (green) (Thermo Fisher Scientific Inc., Waltham, Massachusetts, USA) in 24-well plate for 30 min at 37 °C. Subsequently 1 ul of Hoechst (ThermoFisher, Waltham, Massachusets, USA, cat.n. 33342,) was added to each well for 4 min in room temperature (RT). The caspase positive cells in PIs were analyzed using the same approach as in cell vitality assay mentioned above. Caspase assay was expressed as the percentage of caspase positive cells (apoptotic cells).

**GSIS – glucose stimulated insulin secretion.** Glucose-stimulated insulin secretion was performed in Krebs' solution (prepared by IKEM pharmacy) supplemented with glucose to final concentration 3mM in basal solution (B) and to 22mM concentration in stimulation solution (S). Triplicates from each group composed of 40 PI were placed on inserts with an 8.0 μm membrane (Transwell®,Corning, Somerville, Massachusetts, USA) in 6-well plates and incubated 60 min in basal solution, then in stimulation solution and once again in basal solution in incubator at 37°C. 400 μl samples after each incubation were stored at −20 °C. Insulin levels in all samples were determined using commercially available insulin ELISA kit (Rat insulin Elisa, Mercodia, Uppsala, Sweden) according to manufacturer's protocol. GSIS results were expressed as stimulation index (SI) calculated as the ratio of stimulation value to baseline value.

**Isolation RNA, transcript to cDNA and qRT-PCR gene expression analysis.** The total RNA was isolated with the use of RNeasy Plus Mini Kit (Qiagen, Hilden, Germany) according to the protocol recommended by manufacturer. Total RNA concentration was measured by Qubit™ RNA Broad range (Thermo Fisher Scientific Inc., Waltham, Massachusetts, USA). 0.9–1.6 ug of RNA was used as a template for cDNA. SuperScript™ VILO™ cDNA Synthesis Kit (Thermo Fisher Scientific Inc., Waltham, Massachusetts, USA) was used to manage this procedure.

cDNAs were analyzed by TaqMan qRT-PCR (TaqMan™ Fast Advanced Master Mix for qPCR (Thermo Fisher Scientific Inc., Waltham, Massachusetts, USA)). Gene analysis expressions were performed with *rat* gene specific inflammatory primers IFN-A (Assay ID: Rn02395770_g1), IFN-B (Assay ID: Rn00569434_s1), and TNF-A (Assay ID: Rn99999017_m1) (all from Thermo Fisher Scientific Inc., Waltham, Massachusetts, USA). For reactions and data analysis were used QuantStudio6 Flex System (Thermo Fisher Scientific Inc., Waltham, Massachusetts, USA). Four separate PCRs were performed in doublets, the same amount of cDNA was used for each measurement (49–80 ng cDNA per well). Changes in expression of 3 tested genes were determined using the comparative ΔΔCT method and were normalized to expression of 2 endogenous controls (housekeeping genes), PPIA (Assay ID: Rn00690933_m1) and HPRT (Assay ID: Rn01527840_m1) (both from Thermo Fisher Scientific Inc., Waltham, Massachusetts, USA) [29,30]. Relative expression was calculated using the ΔΔCt method in the relation to the negative control (CTRL-).

**Flow cytometry.** Flow cytometry was performed with PI dissociated into single cells suspension by incubation in Accutase (Merck, Darmstadt, Germany) solution for 20 min at room temperature. The (CTRL-) sample was divided into 5 parts: **unstained control, single stain controls – 3 samples stained with each dye separately and CTRL- sample stained with the complete set of dyes.**

FluoZin™-3 (Thermo Fisher Scientific Inc., Waltham, Massachusetts, USA) was mixed with Pluronic™ F-127 (Thermo Fisher Scientific Inc., Waltham, Massachusetts, USA, cat.n. P3000MP) and diluted with CMRL to final concentration (1,5 µM). Samples were incubated 30 min with occasional mixing in 37 °C, 5% $CO_2$, for incorporation into the cells and binding to $Zn^{2+}$ followed by 30 min incubation after washing under the same conditions for finishing the deesterification.

A volume of 5 µl of APC-conjugated Annexin V (EXBIO Praha, a.s., Vestec, Czech Rep., cat.n. EXB0028) is commonly used per 100 µl of cell suspension. Following centrifugation of the samples at 450 x g for 5 minutes and removal of the supernatant, 100 µl of 1× Annexin V binding buffer (EXBIO Praha, a.s., Vestec, Czech Rep. cat.n. EXB0019) and an appropriate amount of APC-conjugated Annexin V were added to the cell pellet. The samples were vortexed and incubated for 15 minutes at room temperature. Subsequently, the samples were centrifuged again under the same conditions, and the resulting pellets were resuspended in 400 µl of Annexin V binding buffer. To obtain a single-cell suspension, the samples were filtered through a 50 µm filter. Propidium iodide (Merck, Darmstadt, Germany) was added to the samples immediately prior to flow cytometric analysis at a final concentration of 1 µg/ml.

Samples were processed by flow cytometry (BD LSR II, BD Biosciences, San Jose, California, USA) and the data were analyzed using FlowJo software (BD Biosciences, San Jose, California, USA). FluoZin-3–positive beta cells were analyzed to determine the percentage of dead cells (propidium iodide–positive) and apoptotic cells (Annexin V–positive).

## Statistical analysis

Statistical analyses were performed using R software, version 4.4.2.. All data are presented as mean ±SEM. One-way ANOVA following Kruskal-Wallis post hoc test were used for multiple comparisons of data. For statistical evaluation of qRT-PCR data "linear models for microarray data" (LIMMA) was used. All tests were performed as at least 3 independent experiments. As statistically significant were considered values $p < 0.05$.

Significance values were marked as follow: * $p \leq 0.05$, ** $p \leq 0.01$, *** $p \leq 0.001$.

## Results and discussion

The objective of this study was to evaluate the possible impairment of pancreatic islets following transfection with tissue factor targeted siRNA delivered via microporation. Five experimental groups were designed to test all potential detrimental factors. The microporation was performed after overnight stabilization of islets in tissue culture and the testing of its effect was measured 24 hours post-transfection.

The first group served as a negative control (denoted as "CTRL−"), consisting of untreated islets. In the second group ("Si"), siRNA was added directly to the islet culture without microporation. These two groups showed no significant differences across any of the evaluated parameters, confirming that siRNA alone is not passively internalized by islet cells and does not induce cytotoxic effects.

The next two experimental groups consisted of islets exposed to microporation alone ("Mi") or in combination with siRNA ("Mi + Si"). Both conditions showed statistically significant changes in some assays compared to the untreated control group, but with no significant differences observed between them. Taken together, the data suggest that microporation induces a baseline level of islet injury, whereas siRNA delivery does not introduce additional cytotoxicity.

Current papers show that siRNA-based therapies have already received FDA approval for the treatment of specific liver diseases. Additionally, ongoing clinical trials investigate their potential use in addressing conditions affecting the liver, eyes, and skin. Several of these therapies have advanced to Phase 2 clinical trials, with some even reaching Phase 3 [31].

 

The final group, marked as the positive control ("CTRL+"), consisted of islets microporated with Poly(I:C), inducer of cellular stress. Significant impairment observed in the majority of assessed parameters compared to all other groups, it is consistent with the fact that Poly(I:C) is a known activator of inflammation [32]. It is a double-stranded RNA-like structure composed of polyinosinic and polycytidylic acid chains. It mimics viral infection and its structure is recognized by pattern recognition receptors, particularly Toll-like receptor 3 (TLR3), which initiates a signaling cascade leading to the production interferons and proinflammatory cytokines involved in antiviral defense [32–34].

## Light microscopy

Light microscopy images are shown in **Fig 1**, Column 4x, in detail in Column 8x. Healthy islets with smooth, round borders are visible in the CTRL- and Si groups. In contrast, microporated islets from the Mi and Mi + Si groups exhibit disrupted

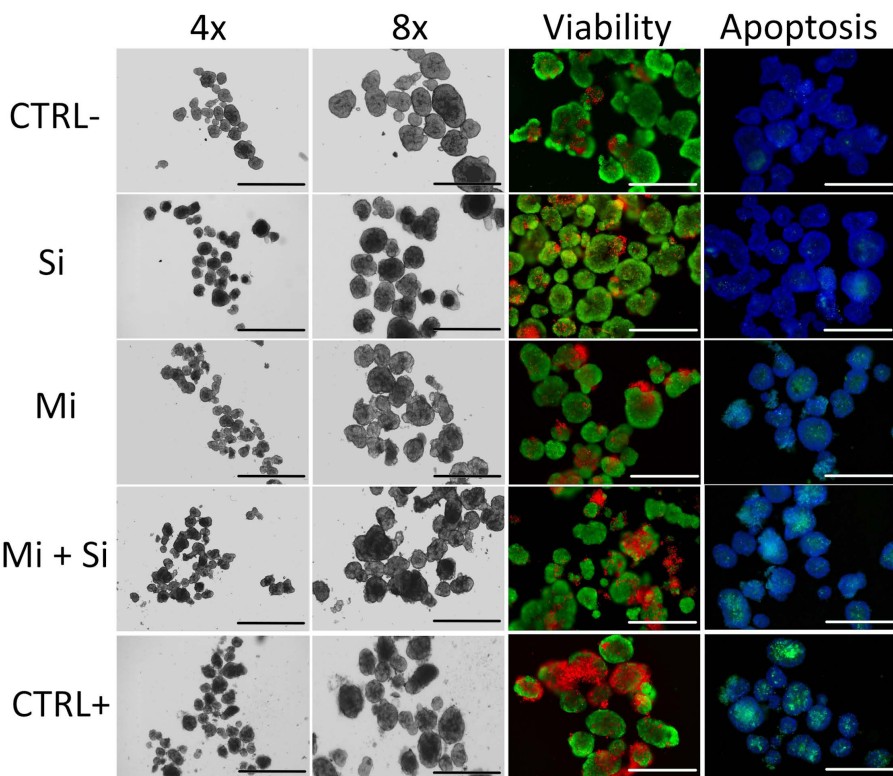

**Fig 1. Light and fluorescent microscopy examination of treated and control islets.** Column 4x- Light microscopy of Pancreatic islets: In the pictures CTRL- and Si are rounded or oval, intact and unglued pancreatic islets. Pictures Mi, Mi + Si and CTRL+ show gradual reduction of ovality and free cells appearing on the surface of islets and even in surrounding of islets. Scale bars in the pictures are 1 000 µm. Column 8x- Detailed view of the image in column A: Cropped section of the first picture (8x) Scale bars are 500 µm. Column Viability – Viability of Pancreatic islets – Propidium iodide and acridine orange staining: In the pictures red color represents dead cells and green color represents live cells in the pancreatic islets. There is evident gradual decreasing viability with increasing number of dead cells in pancreatic islets in order CTRL- to CTRL+. The pancreatic islets in figures CTRL- and Si have a comparable viability, the same applies to Mi and Mi + Si. The worst viability can be seen in the picture CTRL+. Scale bars are 400 µm. Column Apoptosis – Caspase 3/7 staining (apoptosis testing) in Pancreatic islets: The blue color in the image indicates islet cells, with nuclear staining performed using Hoechst 3342, while the green color indicates apoptotic cells positive for caspase −3/7. In the images we can observe an increasing amount of green dye which means caspase 3/7 positive cells in PIs from CTRL- to CTRL+. A recurring pattern, consistent with that observed in the preceding columns, can once again be identified. The CTRL− and Si groups exhibit comparable levels and the Mi and Mi + Si groups show also similarly aligned results to each other. The CTRL+ condition demonstrates the highest number of caspase-positive cells in the PIs. Scale bars are 400 µm. In all pictures: (CTRL- (Negative control – untreated control), Si (siRNA without microporation), Mi (Microporation without siRNA), Mi + Si (Microporation with siRNA), CTRL+ (Positive control – Microporation with Poly(I:C)).

borders, with free, detached cells surrounding the islets. The most extensive damage was observed in the CTRL+ group. The morphological differences observed between healthy and damaged pancreatic islets in this study are consistent with previously published data. Studies comparing morphology of untreated islets with those exposed to proinflammatory cytokines demonstrate similar structural alterations as those seen in our experimental images [35,36].

**Propidium iodide and Acridine orange staining.** Viability staining using Acridine orange and Propidium iodide is a widely accepted method for evaluating islet cells viability. Acridine orange permeates all cells and stains them green while Propidium iodide only enters cells with compromised membranes, staining their nuclei red and thus identifying dead or late apoptotic cells. The mean percentage of living cells in the islets (**Fig 2A**) were as follow: CTRL- (90.43±0.85%), Si (90.48±1.61%), Mi (86.14±1.23%), Mi+Si (84.33±1.22%) and CTRL+ (75.42±11.2%).

Fluorescence staining shows gradually decreasing viability of pancreatic islets from negative to positive control (**Fig 1, column Viability**). This is consistent with the quantitative data obtained from 4 independent experiments evaluated by 3 independent persons.

The high differences in viability were observed in the CTRL+ group, which showed a high level of statistical significance (p<0.001) compared to the CTRL-, Si and Mi+Si groups. At the same time, compared to the Mi group, the CTRL+ group showed statistical significance at the p<0.001 level. Moreover, the difference between CTRL+ and Mi+Si group reached statistical significance at the p=0.001 level. Additionally, moderate significance was also detected between the Mi+Si group and both the CTRL- (p=0,025) and Si (p=0,024) groups. These findings confirmed that the viability of the CTRL- and Si samples is nearly identical, approximately 90.4%, which aligns with the previously reported typical viability of *rat* pancreatic islets cultured for 1–2 days [37,38]. The viability of microporated islets, with or without siRNA, showed only approximately 5% lower viability compared to the control untreated and siRNA-only samples, suggesting that microporation introduces only mild cellular stress. Our results are in agreement with previously published results on other cell types, such as mesenchymal stromal cells (MSC) which were microporated with pEGFP-N1 vector and exhibited viability around 80% 48 hours post-microporation [18]. The lowest viability among all tested samples was observed in the positive control sample (CTRL+) confirming damaging impact of Poly(I:C) [32], which triggers an inflammatory response in pancreatic islets.

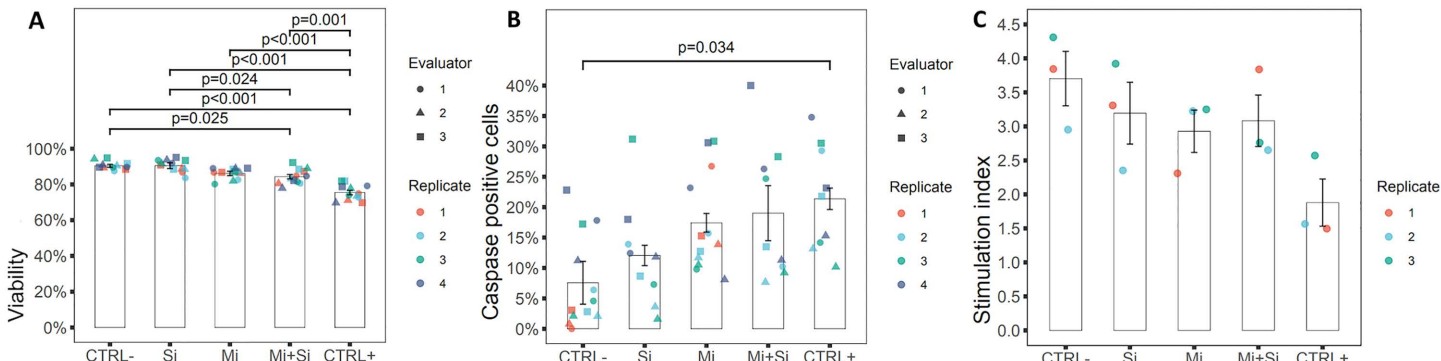

**Fig 2. Pancreatic islets viability, apoptosis and insulin secretion capacity after microporation of siRNA. A.** Viability – Viability was assessed by three independent evaluators based on images shown in Fig 1, column Viability. The contribution of each evaluator is represented by different symbols, while individual replicates are distinguished by colors. A gradual decrease in viability from the negative to the positive control can be observed. **B.** Apoptotic cells – Apoptotic cells in pancreatic islets were evaluated in the same manner as viability. A statistically significant difference was observed only between the negative and positive controls. Nevertheless, a gradual increase in the number of apoptotic cells from the negative to the positive control can also be observed. **C.** Insulin secretion: Glucose stimulated insulin secretion after treatment of Pancreatic islets expressed by the stimulation index. Individual replicates are indicated by colored dots. No statistically significant differences were observed. In all pictures: CTRL- (Negative control – untreated control), Si (siRNA without microporation), Mi (Microporation without siRNA), Mi+Si (Microporation with siRNA), CTRL+ (Positive control – Microporation with Poly(I:C).

**Activation of caspase 3/7 by siRNA.** In this study, caspase 3/7 staining (green fluorescence) was utilized as a marker of programmed cell death within pancreatic islets, while Hoechst dye was used to visualize cell nuclei. As shown in Fig 1 column Apoptosis, caspase activity remained low and non-significant difference between untreated control islets (CTRL−) and in islets treated with siRNA alone (Si). In contrast, an increase in apoptotic cell number was observed following microporation (Mi), and a slight additional increase was seen when microporation was combined with siRNA delivery (Mi + Si). The strongest apoptotic response was detected in the positive control group (CTRL+), which underwent microporation with Poly (I:C), an activator of innate immune pathways.

These qualitative observations were supported by quantitative analysis performed by three independent evaluators across four separate experiments. The proportion of caspase 3/7-positive cells within islets is presented in Fig 2B and was as follows: CTRL− (7.56 ± 3.51%), Si (12.05 ± 1.68%), Mi (17.42 ± 1.54%), Mi + Si (19.01 ± 4.53%), and CTRL+ (21.37 ± 1.77%). Among all groups, the only statistically significant difference (p = 0,034) was observed between the negative and positive control groups, confirming that Poly (I:C)-induced apoptosis involves activation of the caspase pathway. This result is in line with previously reported findings [39] demonstrated that Poly (I:C) induces caspase-mediated apoptosis in human CD34 + cells.

**Glucose stimulated insulin secretion GSIS was not impaired by microporation of siRNA.** Microporation of siRNA does not impair glucose-stimulated insulin secretion (GSIS). The stimulation index (SI) values for the Si (3.19 ± 0.45), Mi (2.93 ± 0.31), and Mi + Si (3.08 ± 0.38) groups were slightly lower than that of the untreated control islets CTRL- (3.7 ± 0.40), and this decrease is clearly visible in the Fig 2C. In contrast, a reduction in SI was observed in the CTRL+ group (SI = 1.88 ± 0.35); however, none of the differences among experimental groups reached statistical significance (p > 0.05). The experiment was independently replicated three times. While current data suggest a lack of significant functional impairment, additional replicates may be required to detect subtle effects.

Our observations are in line with previous reports. Lefebvre et al. [40] demonstrated that microporation of an eGFP-expressing plasmid, following mild enzymatic dissociation with Accutase, did not significantly alter insulin secretion. Similarly, both Weber et al. [41] and DeLeu et al. [42] reported that adenoviral transduction with ß-galactosidase nor mRNA for VEGF, respectively had not any significant impact on insulin secretion. Together, these studies and our data support the conclusion that microporation-based gene delivery preserves functional integrity of pancreatic islets.

**qRT-PCR of Inflammatory markers.** siRNAs have been reported to trigger innate immune responses. Both double-stranded and single-stranded siRNAs can induce an antiviral immune reaction, leading to the secretion of number of various cytokines [43–45]. Most cell types react to a viral infection by secreting interferons IFN-α/β. In contrast, the production of IFN-γ and TNF-α is mainly restricted to immune cells. These include natural killer cells, T- lymphocytes, dendritic cells or macrophages [46,47]. Considering these facts, the possible risk of side effects caused by the specific synthetic siRNA used in this study was tested.

For our experiments, we used poly(I:C) as a positive control for the activation of innate immunity. Poly(I:C) is a double-stranded RNA (dsRNA). It is often used as a potent adjuvant to enhance the efficacy of vaccines against viruses. Poly(I:C) acts by triggering antiviral pathways and, among other things, leads to an increase in IFN-β [48].

Across all genes examined, the most substantial differences in expression relative to the negative control (CTRL−) was consistently detected in the positive control group (CTRL+), subjected to microporation with Poly(I:C). This indicates a robust activation of the proinflammatory pathways by universal positive control.

Expression of CTRL+ was increased 5.6-fold for IFNα (**Fig 3A**), 205-fold for IFNβ (**Fig 3B**) and 34-fold for TNFα compared to CTRL – (**Fig 3C**), with all changes being statistically significant. Moreover, the CTRL+ group differ significantly from all experimental groups across all tested genes. Although, the degree of significance varied (**Fig 3**).

In contrast, all other experimental groups showed only mild increase in IFN α expression relative to CTRL- without reaching of statistical significance. The expression of IFN α (**Fig 3A**), was 1.12-fold higher for the Si group, 1.25-fold higher for the Mi group, and 1.24-fold higher for the Mi + Si group.

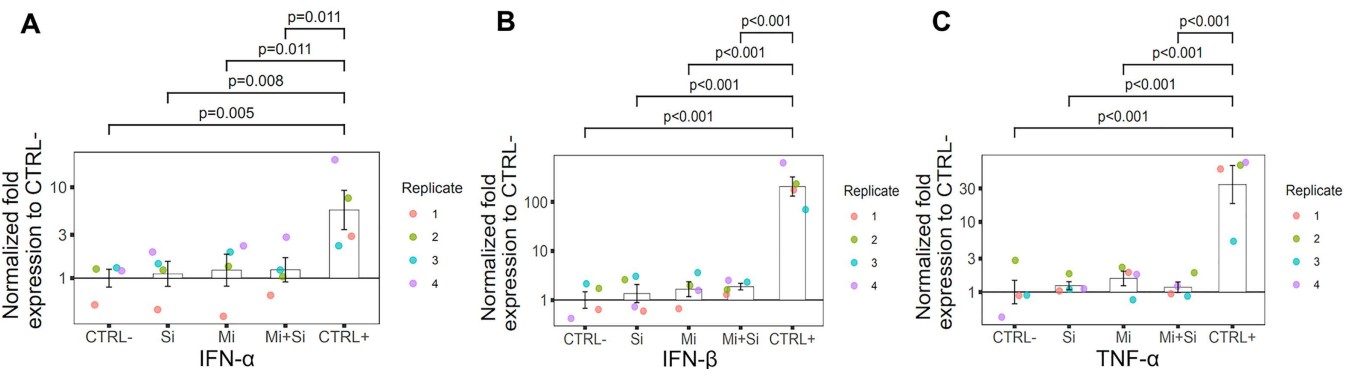

**Fig 3. The expression of selected inflammatory genes in treated and control islets.** Results of inflammatory markers measured by qRT-PCR in pancreatic islets with and without siRNA and microporation treatment. IFN α, IFN β and TNF α markers were measured. The figures show a significant change in the expression of all measured markers just for the positive control (pancreatic islets microporated with Poly(I:C). Individual replicates in graphs are color-coded. In all pictures: CTRL- (Negative control – untreated control), Si (siRNA without microporation), Mi (Microporation without siRNA), Mi+Si (Microporation with siRNA), CTRL+ (Positive control – Microporation with Poly(I:C).

The expression of IFN β (**Fig 3B**) was for Si sample 1.36- fold higher, for Mi sample 1.6- fold higher, and for Mi+Si sample 1.8- fold higher. For both interferons, there was a slight increase in the expression rate in the sequential order of Si, Mi and Mi+Si. However, these results were not statistically significant. For TNFα (**Fig 3C**), the rest of the results showed the largest change in expression compared to the CTRL- for the Mi sample, 1.56-fold (excluding CTRL+). This was followed by the Si sample with a 1.23-fold higher expression. In this case, the Mi+Si sample had the lowest difference in expression rate, 1.17-fold. Still, none of these results were statistically significant.

Taken together, qRT-PCR analysis revealed that all 3 approaches (Si, Mi and Mi+Si) induced only minimal immune response, while microporation with Poly(I:C), CTRL+, triggered rapid upregulation of all tested proinflammatory markers. These results are consistent with available literature reporting that Poly(I:C) induces strong activation of immunoregulatory genes [49,50]. Reimer et al. [49] conclude their experiments that Poly(I:C) induces an overlapping cytokine response, especially induce IFN beta, which is in agreement with our results. Poly(I:C) innate immune activation is mediated mainly through a number of receptors known as Toll-like receptors [48]. The receptors are expressed on cells of the immune system including neutrophils [50,51]. However, it has been recognized that they are also expressed on other cells as well as on islet cells [51,52]. This may explain why immune activation occurs in isolated pancreatic islets after Poly(I:C) transfection. The results proved, that Poly(I:C) can be used as an optimal positive control for the monitoring of the cells immune response. On the contrary, qRT-PCR showed that the microporation is a suitable and non-immunogenic method for delivering siRNA into pancreatic islets.

**Flow cytometry.** Flow cytometry offers a quick and easy way to identify and quantify alive, apoptotic, and dead cells. For this purpose, we used Accutase, a gentle enzymatic agent previously utilized for islet cell dissociation [53,54]. However, enzymatic disintegration of islets into a single-cell suspension can negatively affect cell viability [55,56], especially for islets treated with invasive procedures like microporation. This issue was solved by experiment design including suitable control groups.

Among the islet cells, we identified beta cells using FluoZin-3, a fluorescent indicator for $Zn^{2+}$ and therefore a vital beta cell dye [57] (**Fig 4**). To detect early-stage apoptosis, Annexin V is a commonly used marker [58]. Since Annexin V can enter necrotic cells through injured membranes, identification of apoptotic cells requires simultaneous negativity necrotic (dead) cells [59,60]. In our study, Propidium iodide was used for this purpose. Thus, in flow cytometric analysis, cells positive for Annexin V and negative for Propidium iodide were considered apoptotic (**Fig 4**).

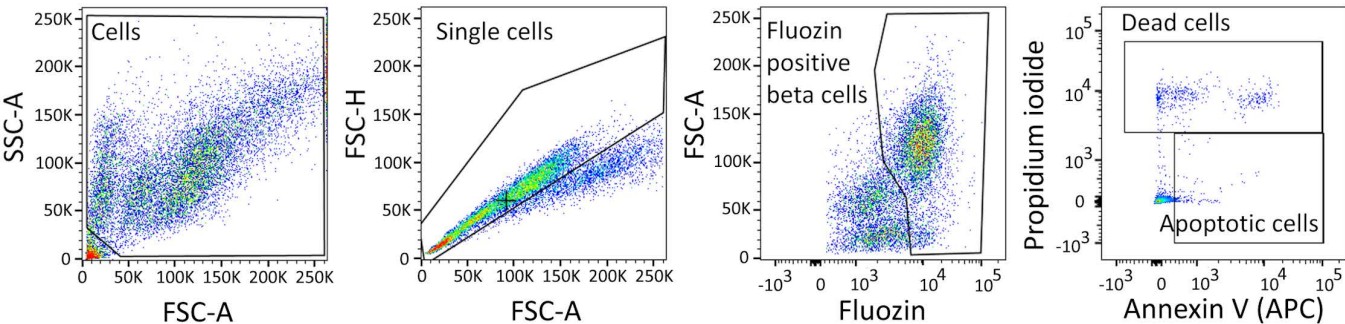

**Fig 4. Gating strategy for flow cytometric analysis of beta cell viability.** Rat islet samples were enzymatically dissociated into cell suspension and labeled for flow cytometric analysis. Sequential gating was used for each islet sample to obtain populations of cells, single cells, and FluoZin-3 positive beta cells. Beta cells were analyzed for percentage of apoptotic cells (Annexin V positive cells) and dead cells (Propidium iodide positive cells).

The CTRL- and Si groups showed the lowest percentages of Annexin V positive cells (2.37% and 2.63%, respectively). The Mi and Mi+Si groups had significantly more apoptotic cells (6.96% and 5.48%, respectively) compared to both the CTRL- (p<0,001 and p=0,003) and Si (p<0,001 and p=0,006) groups. The two groups with microporated islets (Mi and Mi+Si) did not differ significantly in the degree of islet cell apoptosis from each other. The highest percentage of apoptosis was observed in the CTRL+ group (7.88%), which was significantly different from the non-microporated groups (CTRL-, Si for both p<0,001) and the Mi+Si (p=0,018) group. However, the CTRL+ group did not differ from the Mi group, suggesting that microporation with Poly(I:C) did not increase apoptosis levels compared to microporation alone (**Figs 5** and **6**).

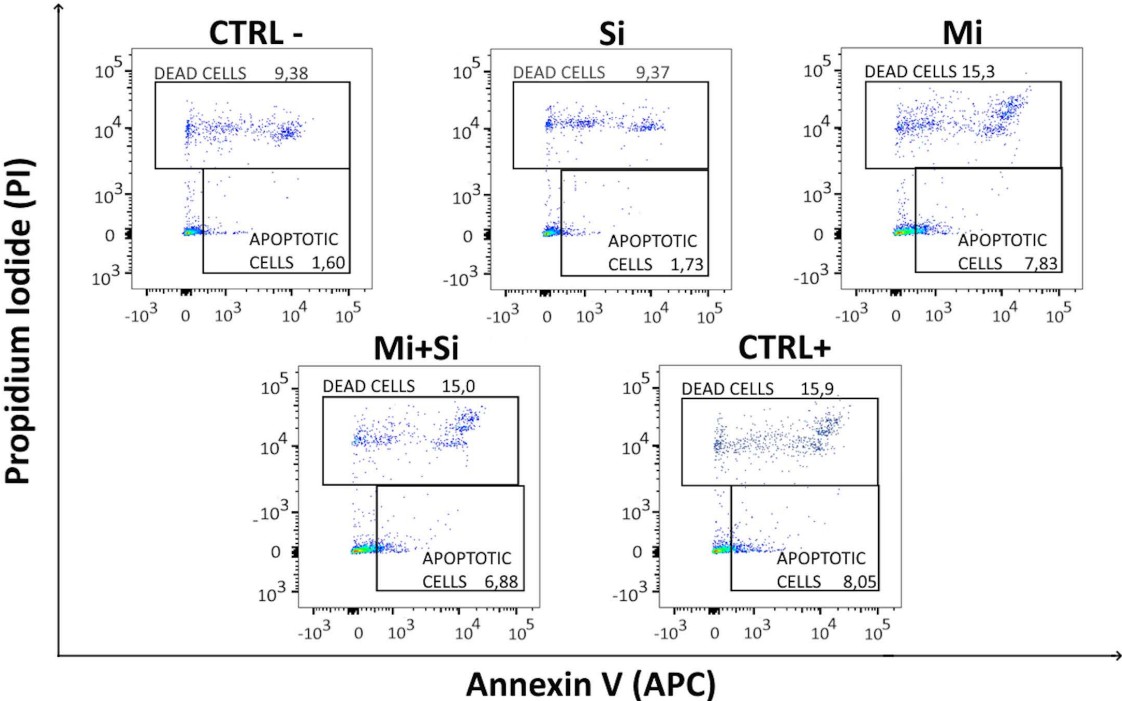

**Fig 5. Flow cytometric detection of cell viability.** The representative plots of dead and apoptotic beta cells of treated Pancreatic islets stained with Annexin V and Propidium iodide: CTRL- (Negative control – untreated control), Si (siRNA without microporation), Mi (Microporation without siRNA), Mi+Si (Microporation with siRNA), CTRL+ (Positive control – Microporation with Poly(I:C)).

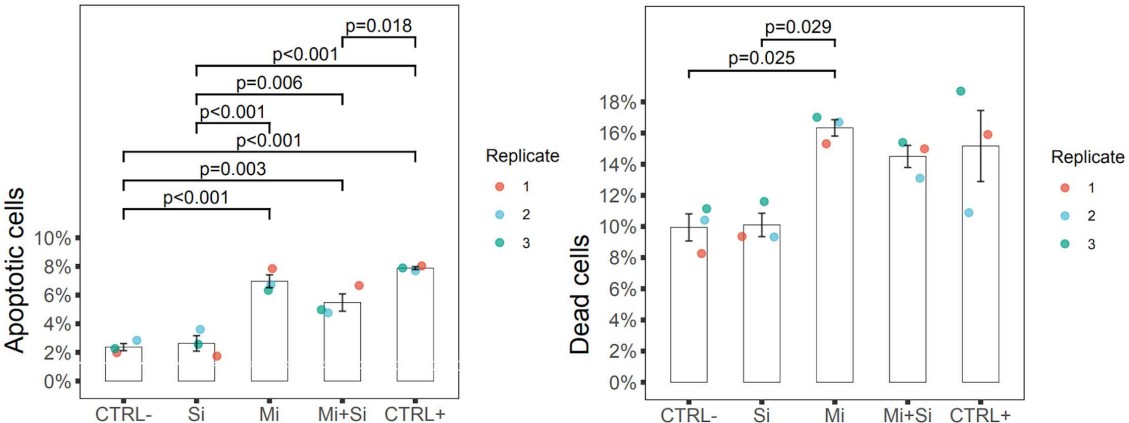

**Fig 6. Flow cytometric quantitative analysis of cell viability.** The graphs presenting percentage of apoptotic and dead beta cells of treated Pancreatic islets. In both graphs individual replicates are distinguished by different colors. Islets that were not subjected to microporation exhibited lower percentages of both dead and apoptotic cells within the islets. In all pictures: CTRL- (Negative control – untreated control), Si (siRNA without microporation), Mi (Microporation without siRNA), Mi+Si (Microporation with siRNA), CTRL+ (Positive control – Microporation with Poly(I:C).

The relatively low percentage of apoptotic islet cells, even in the CTRL+ group, might be due to the late timing of analysis after microporation. Cells stimulated by microporation to undergo apoptosis might have already permeabilized their membranes to Propidium iodide and transformed into dead cells. Annexin V detects the translocation of phosphatidylserine to the outer surface of the plasma membrane, an early event in apoptosis that precedes membrane permeabilization, cell shrinkage, and nuclear condensation. This exposure triggers recognition and engulfment of apoptotic cells by phagocytes [61,62]. The early phase of apoptosis, detectable by Annexin V, can be initiated within minutes to a few hours after an apoptotic stimulus [63,64]. In the absence of phagocytosis, apoptotic cells progress to secondary necrosis, characterized by features similar to necrotic cells [65,66]. This can be observed hours after the apoptotic stimulus [65,67]. The exact timing varies between cell types and is not known for beta cells.

This aligns with the comparison to the rate of islet cell positivity for active caspase 3. Caspase 3 is activated early in apoptosis, and its activity persists into later stages, triggering the DNA degradation, cytoskeleton remodelling, and cell death [68]. Therefore, cells positive for activated caspase 3 can be identified as dead in cytometric analysis (Propidium iodide positive, Annexin V positive) despite being in a late stage of apoptosis. The results from both methods are in agreement because the sum of Annexin positive cells and dead cells corresponds to the number of caspase 3/7 positive cells. Thus, the Annexin V positivity of PI cells in our study might represent the susceptibility of these cells to dissociation, a susceptibility that could also be influenced by prior microporation and siRNA treatment.

The percentage of dead cells (Propidium iodide positive) was lowest in the non-microporated groups (CTRL- 9.94% and Si 10.10%), with no significant difference between them. A significant increase in dead cells compared to non-microporated islets (CTRL-, Si) was only observed in the Mi group (16.33% dead cells; CTRL- vs Mi p=0,025 and Si vs Mi p=0,029). However, the Mi and Mi+Si groups did not significantly differ in the number of dead cells (16.33% and 14.5%). The CTRL+ group also did not differ from other groups in the number of dead cells (15.17%). These data indicate that siRNA treatment does not increase the percentage of dead cells in islets (**Figs 5** and **6**).

The number of dead cells identified by flow cytometry corresponds with viability staining using Acridine orange and Propidium iodide. The lower number of dead cells in the CTRL+ group by flow cytometry (15.17% vs 24.58%) could be due to complete disintegration of cells during islet dissociation, as cells affected by Poly(I:C) might be more susceptible. The significantly lower amount of apoptotic and dead cells in the Mi+Si group compared to the CTRL+ group in flow

cytometry analysis could suggest a slightly positive effect of siRNA. However, the effect of siRNA was not sufficient to reduce the rate of apoptosis and islet cell death in the Mi + Si group compared to the Mi group.

## Conclusion

In summary, our findings indicate that neither microporation nor the siRNA applied in this study adversely affect pancreatic islet viability and function. The siRNA itself is non-cytotoxic and is not passively taken up by islet cells. Although microporation induces measurable but not significant cellular stress, the combination with siRNA does not aggravate this effect. These results support the conclusion that microporation is a well-tolerated method for nucleic acid delivery, offering a safe and effective approach for targeted gene silencing in pancreatic islet cells. These findings can motivate other research groups to use this elegant method of temporarily silencing the expression of selected genes without the risk of affecting nuclear DNA reliably employed in future experimental studies for treating some patients with type 1 diabetes mellitus.

## Acknowledgments

The authors would like to thank Eva Dovolilová for her reliable and helpful assistance in performing insulin secretion tests, Magdalena Špitálníková Veřtátová for her invaluable help in performing cytometric examinations and processing all results, and Ing. István Módos, Ph.D. for statistical analysis of data obtained.

## Author contributions

**Conceptualization:** Klara Zacharovova, Eva Fabryova, Tomas Koblas, Zuzana Berkova.

**Data curation:** Veronika Tomsovska, Ivan Leontovyc, Klara Zacharovova, Eva Fabryova, Zuzana Berkova.

**Formal analysis:** Veronika Tomsovska, Klara Zacharovova, Eva Fabryova, Tomas Koblas.

**Funding acquisition:** Jan Kriz.

**Investigation:** Veronika Tomsovska, Zuzana Berkova.

**Methodology:** Veronika Tomsovska, Ivan Leontovyc, Tomas Koblas.

**Project administration:** Eva Fabryova.

**Resources:** Jan Kriz.

**Supervision:** Jan Kriz.

**Writing – original draft:** Veronika Tomsovska, Klara Zacharovova, Tomas Koblas.

**Writing – review & editing:** Zuzana Berkova, Jan Kriz.

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
