## [Decision Letter · Decision Letter 0]

26 Aug 2025

Dear Dr. Kriz,

Thank you for submitting your manuscript to PLOS ONE. After careful consideration, we feel that it has merit but does not fully meet PLOS ONE’s publication criteria as it currently stands. Therefore, we invite you to submit a revised version of the manuscript that addresses the points raised during the review process.

We look forward to receiving your revised manuscript.

Kind regards,

Kota V Ramana, Ph.D.

Academic Editor

PLOS ONE

Journal Requirements:

“1) Supported by the project National Institute for Research of Metabolic and Cardiovascular Diseases (Programme EXCELES, ID Project No. LX22NPO5104) - Funded by the European Union – Next Generation EU.

2) Supported by Ministry of Health, Czech Republic - conceptual development of research organization (“Institute for Clinical and Experimental Medicine – IKEM, IN 00023001“)

3) “Supported by the Ministry of Health of the Czech Republic in cooperation with the Czech Health Research Council under project No. NW25-01-00221. All rights reserved."”

Reviewers' comments:

Reviewer's Responses to Questions

**Comments to the Author**

1. Is the manuscript technically sound, and do the data support the conclusions?

Reviewer #1: Partly

Reviewer #2: Yes

2. Has the statistical analysis been performed appropriately and rigorously?

Reviewer #1: No

Reviewer #2: Yes

3. Have the authors made all data underlying the findings in their manuscript fully available?

Reviewer #1: Yes

Reviewer #2: Yes

4. Is the manuscript presented in an intelligible fashion and written in standard English?

Reviewer #1: Yes

Reviewer #2: No

Reviewer #1: 1. Please indicate to the aim of the study in introduction section, remove the last 3 sentences about what you did.

2. Use the same font for title and text.

3. In method section, provide explanation about the methods which used by detail and write continuously in one paragraph for each method, not just mention as i. ii, etc.

4. Indicate to the cell number which used for transfection.

5. Indicate to the house keeping gene for q-RT-PCR evaluation.

6. Indicate to the software you used for statistical analysis.

7. Indicate to the exact value of p value in results.

8. The conclusion section contain redundancy, rewrite this part.

9. In conclusion section, you should just indicate to the main findings and decision you got of the study. You should not indicate to the references .

10. The lack of comparison between your findings and previous studies has been found in discussion section.

11. Indicate to the primer sequences in the table.

12. Most of the references are out of date, try to use references in recent 10 years.

Reviewer #2: Summary

This manuscript is overall well written and presents relevant findings on the innocuous effects microporation for transfecting pancreatic islets with siRNA for a possibility to improve transplantation success in diabetic patients; however, the figures, figure captions, and data visualization require revisions before the study can be accepted for publication. Clearer plots, consistent annotations, and improved captions will greatly enhance readability and strengthen the presentation of results.

Major Comments

Figure 1

- Update the titles of columns A and B to indicate the magnification used for image acquisition (e.g., 10×, 20×).

- Scale bars: Ensure that scale bars are visible, consistent, and clearly labeled across all panels. The grey scale bar in column A is barely noticeable, and column B appears to lack one. For columns C and D, the small text below the scale bars is unreadable; I recommend removing this text and instead reporting scale bar lengths in the figure caption.

- Rename the titles of columns C and D to something more descriptive, such as “Viability (PI/DAPI?)” and “Apoptosis (Caspase3/7)”.

- Color coding: Specify what each color represents (e.g., green, red, blue) for the stains used. This information should be provided either directly in the figure legend or in the caption for clarity.

- Optimize layout: Reduce the excessive spacing between images to improve figure readability.

Figure 2

- Explicitly state on the figure caption what the black dots and vertical lines represent in the plots.

- Replot the data using transparent boxplots to show group averages, and overlay individual replicate values as jittered points for clarity.

- High-resolution data representation: The manuscript states that quantitative analyses were performed by three independent evaluators across four separate experiments. This rich dataset could be presented more effectively by including on the plots:

- Individual replicates

- Evaluator contributions

- Group averages

- Statistical test results

- Guidelines for visualization: I recommend consulting Lin & Landry (2024; https://www.biorxiv.org/content/10.1101/2024.09.20.609464v2) for best practices in boxplot data visualization.

- Formatting adjustments: Increase the size of axis titles, tick labels, and asterisks indicating statistical significance to improve readability.

Figure Captions (General)

The figure captions are currently difficult to follow and should be restructured for greater clarity and consistency:

- Group related panels logically: For example, in Fig. 1, combine columns A+B as Fig. 1A, column C as Fig. 1B, and column D as Fig. 1C.

- Focus on describing the content: Captions should state what is being shown (e.g., variables, treatments, measurements, stats) rather than primarily summarizing results. If desired, key findings can be highlighted in the figure title instead.

- Include statistical details: Report relevant information (e.g., tests performed, sample sizes, p-values) within the captions themselves, not only in the Methods section.

- I recommend reviewing Jambor et al. (2021; https://journals.plos.org/plosbiology/article?id=10.1371/journal.pbio.3001161) and Liu et al. (2023; https://www.sciencedirect.com/science/article/abs/pii/S0889490622000606) for figure legend description.

Figure 3

- Apply the same recommendations as for Figure 2:

- Use transparent boxplots with jittered replicates.

- Increase font sizes for axis titles, tick labels, and statistical annotations.

Figures 4–5

- If possible, increase the font size for all labels and annotations to improve readability.

Figure 6

- Replot the data using the same recommendations as for Figures 2 and 3:

- Transparent boxplots

- Jittered replicates

- Inclusion of statistical information and additional data

Minor Comments

- Line 89: Correct “Ratus norvergicus” → Rattus norvegicus (capitalize R).

- Line 274: Rephrase for clarity: “Poly(I:C) is a known activator of inflammation.”

- Line 533: Update phrasing using “on”, “…caused any negative effects on pancreatic islets…”.

- Line 535: Improve the ending of the phrase by giving the research overall purpose: “…reliably employed in future experimental studies for treating some patients with type 1 diabetes mellitus.”

The manuscript presents valuable findings on siRNA as a prospective method to improve the survival of transplanted pancreatic islets, but the figures and captions require revisions to enhance clarity, readability, and consistency. Improving data visualization and providing comprehensive, well-structured figure legends will significantly strengthen the overall impact and interpretability of the study.

**Do you want your identity to be public for this peer review?** For information about this choice, including consent withdrawal, please see our Privacy Policy

Reviewer #1: **Yes:** Saiedeh Razi-Soofiyani

Reviewer #2: No

---

## [Author Response · Author response to Decision Letter 1]

16 Oct 2025

Our responses to reviewer’s comments and requests are as follows:

Reviewer #1:

1. Please indicate to the aim of the study in introduction section, remove the last 3 sentences about what you did.

The last paragraph of the Introduction was rephrased as requested (lines 87-95 in tracked changes file).

2. Use the same font for title and text.

The fonts were part of the Word settings, and their purpose was to highlight individual heading levels. I'm not sure if this is a significant error, given that the publisher will modify all fonts for printing anyway to match the journal layout, but we unified the fonts according to the reviewer's request.

3. In method section, provide explanation about the methods which used by detail and write continuously in one paragraph foreach method, not just mention as i. ii, etc.

The method Section: „Evaluation of pancreatic islets after treatment“ contains a list of methods used under letters i., ii., etc. for clarity. The individual methods are described in detail in the following text, as requested by the reviewer.

4. Indicate to the cell number which used for transfection.

The text was rephrased as requested (lines 137, 146 in tracked changes file).

5. Indicate to the house keeping gene for q-RT-PCR evaluation.

The text was rephrased as requested (lines 213 in tracked changes file).

6. Indicate to the software you used for statistical analysis.

The text was rephrased as requested (lines 248, 249 in tracked changes file).

7. Indicate to the exact value of p value in results.

The text was rephrased as requested (lines 337-341, 366, 466, 467, 470, 471, 513-514 in tracked changes file).

P-values: If any statistical significance was stronger than p < 0.001, the exact number is not written by the software.

8. The conclusion section contain redundancy, rewrite this part.

The text of Conclusion section was rephrased as requested. The redundant information was erased and the suggested contribution to other groups was added (lines 535 - 545).

9. In conclusion section, you should just indicate to the main findings and decision you got of the study. You should not indicate to the references.

The text of Conclusion section was rephrased as requested. The redundant information was erased and the suggested contribution to other groups was added (lines 535 - 545).

10. The lack of comparison between your findings and previous studies has been found in discussion section.

Our feeling is the same, but the problem is, that there are not several studies using microporation for transfection of molecules to pancreatic islet cells. The comparison to basic electroporation or lipofection is not reliable. But we tried to do our best and add some references to the text of “Results and discussion” section as follows:

Line 382 the Lefebvre paper (40) is the newest paper of this problematic we have found

Line 384 the reference of Nico DeLeu 2014 was added and the sentence slightly rephrased

Line 397 the reference of Garcia-Sastre 2017 was added

Line 439, 440 the reference of El-Zayat 2019 et al was added

Line 450 the reference of Lee 2020 was added

11. Indicate to the primer sequences in the table.

All primers TaqMan qRT PCR analysis are mentioned in methods section with specific ID in catalogue of ThermoFisher® Scientific company. The sequences were not provided by the supplier as it is a protected know-how of the company. The genomic maps of target mRNA and symbolic location of the primer were provided.

12. Most of the references are out of date, try to use references in recent 10 years.

We added several current references as requested

Reviewer #2 - major comments:

Figure 1 - The picture was edited according to request of the reviewer. Description of scale bars was moved from the picture to the picture legenda, titles were renamed and colors are explanted in the picture description.

Figure 2 - The graph and its description were modified by the request of the reviewer 2. The statistician modified it as much as possible to bar plots in order to keep it correct.

Figure Captions (General) - Thank you for this comment. We have standardized the graph labels and focused more consistently on describing the information displayed in the graphs with an indication of the intervention used. The font is significantly larger to make it easier to read. We hope, the reviewer would feel it improved.

Figure 3 - The graph and its description were modified by the request of the reviewer. The statistician modified it as much as possible to bar plots in order to keep it correct.

Figures 4–5 - Was done as requested

Figure 6 - The graph and its description were modified by the request of the reviewer. The statistician modified it as much as possible to bar plots in order to keep it correct.

Reviewer #2 - minor comments:

• Line 89: Correct “Ratus norvergicus” → Rattus norvegicus (capitalize R).

Done

• Line 274: Rephrase for clarity: “Poly(I:C) is a known activator of inflammation.”

Done

• Line 533: Update phrasing using “on”, “…caused any negative effects on pancreatic islets…”.

The sentence was completely replaced according to request of the reviewer 1.

• Line 535: Improve the ending of the phrase by giving the research overall purpose: “…reliably employed in future experimental studies for treating some patients with type 1 diabetes mellitus.”

Done

---

## [Decision Letter · Decision Letter 1]

16 Dec 2025

Evaluation of rat pancreatic islets damage after siRNA microporation

PONE-D-25-41672R1

Dear Dr. Kriz,

We’re pleased to inform you that your manuscript has been judged scientifically suitable for publication and will be formally accepted for publication once it meets all outstanding technical requirements.

Kind regards,

Kota V Ramana, Ph.D.

Academic Editor

PLOS One

Additional Editor Comments (optional):

Reviewers' comments:

Reviewer's Responses to Questions

**Comments to the Author**

Reviewer #1: All comments have been addressed

2. Is the manuscript technically sound, and do the data support the conclusions?

Reviewer #1: Yes

3. Has the statistical analysis been performed appropriately and rigorously?

Reviewer #1: Yes

4. Have the authors made all data underlying the findings in their manuscript fully available?

Reviewer #1: Yes

5. Is the manuscript presented in an intelligible fashion and written in standard English?

Reviewer #1: Yes

Reviewer #1: (No Response)

**Do you want your identity to be public for this peer review?** For information about this choice, including consent withdrawal, please see our Privacy Policy

Reviewer #1: No

---

## [Editor Report · Acceptance letter]

PONE-D-25-41672R1

PLOS One

Dear Dr. Kriz,

I'm pleased to inform you that your manuscript has been deemed suitable for publication in PLOS One. Congratulations! Your manuscript is now being handed over to our production team.

Kind regards,

on behalf of

Dr. Kota V Ramana

Academic Editor

PLOS One